# Switchable Electromagnetically Induced Transparency with Toroidal Mode in a Graphene-Loaded All-Dielectric Metasurface

**DOI:** 10.3390/nano10061064

**Published:** 2020-05-30

**Authors:** Guanghou Sun, Sheng Peng, Xuejin Zhang, Yongyuan Zhu

**Affiliations:** 1National Laboratory of Solid State Microstructures, Key Laboratory of Intelligent Optical Sensing and Manipulation, Jiangsu Key Laboratory of Artificial Functional Materials, Collaborative Innovation Center of Advanced Microstructures, School of Physics and College of Engineering and Applied Sciences, Nanjing University, Nanjing 210093, China; ghsun2003@163.com (G.S.); dg1822027@smail.nju.edu.cn (S.P.); yyzhu@nju.edu.cn (Y.Z.); 2School of Science and Key Laboratory of Jiangxi Microstructure Functional Materials, Jiujiang University, Jiujiang 332005, China

**Keywords:** electromagnetically induced transparency, toroidal dipole, dielectric metasurface, graphene, switchable, slow light

## Abstract

Active photonics based on graphene has attracted wide attention for developing tunable and compact optical devices with excellent performances. In this paper, the dynamic manipulation of electromagnetically induced transparency (EIT) with high quality factors (Q-factors) is realized in the optical telecommunication range via the graphene-loaded all-dielectric metasurface. The all-dielectric metasurface is composed of split Si nanocuboids, and high Q-factor EIT resonance stems from the destructive interference between the toroidal dipole resonance and the magnetic dipole resonance. As graphene is integrated on the all-dielectric metasurface, the modulation of the EIT window is realized by tuning the Fermi level of graphene, engendering an appreciable modulation depth of 88%. Moreover, the group velocity can be tuned from *c*/1120 to *c*/3390. Our proposed metasurface has the potential for optical filters, modulators, and switches.

## 1. Introduction

Electromagnetically induced transparency (EIT) originates from a quantum interference in atomic physics, leading to the formation of a narrow transparency window within a broad absorption spectrum [1]. This concept was later extended to metamaterial systems. The EIT effect in metamaterials mainly results from two interference pathways: bright–dark mode coupling and bright–bright mode coupling [2,3]. In the narrow transparency window, the strong dispersion behaviors make it advantageous for applications in optical modulators [4], sensors [5] and slow-light devices [6]. In recent years, the active control of optical responses has been a research hotspot in the metamaterial field, which provides an additional degree of freedom to precisely manipulate the light–matter interaction. Some schemes have been proposed to realize a dynamical control of the EIT effect based on metamaterials, such as liquid crystals [7], photoactive semiconductors [8], phase change media [9], and electrical control [10].

Conventional metallic metamaterials suffer from large intrinsic Ohmic losses. The quality factors (Q-factors) of their plasmonic resonances are extremely low [11], limiting the modulation performance. By comparison, all-dielectric materials offer a potential solution to the issue of intrinsic losses. All-dielectric metasurfaces with multiple functions are realized based on the Mie resonances of dielectric particles, such as flat lenses [12,13], sensors [14], and beam shaping [15]. Various kinds of high Q-factor resonances have been studied in all-dielectric metamaterials [14,16,17,18,19]. On the other hand, graphene has attracted considerable interest because of its unique properties of ultra-high electron mobility and tunable carrier density. Its conductivity can be dynamically modulated by changing the Fermi level through electrostatic gating or chemical doping [20]. Hybrid graphene–dielectric metasurfaces exhibit excellent modulation performance based on Fano resonance [21,22,23]. Nevertheless, they are not suitable for optical switching. As a peculiar electromagnetic excitation, different from electric and magnetic dipoles, the toroidal dipole with electrical currents circulating on the surface of a torus along its meridians, has become a research focus in photonics. The toroidal dipole in natural materials is difficult to observe due to the simultaneous excitation of much stronger electric and magnetic multipoles [24]. Fortunately, the toroidal dipole can be prominently enhanced by plasmonic or all-dielectric metamaterials [25,26]. However, their geometric structures are relatively complex for nanofabrication. It is natural to expect that in simpler structures the toroidal dipole can be excited, giving us a chance to conveniently apply it to various kinds of optical devices. Recently, the toroidal dipole has gained more attention in all-dielectric nanostructures [26,27,28,29,30,31]. By the coupling of toroidal dipole resonance and radiative modes, high Q-factor Fano and EIT resonances can be obtained. Combined with the advantages of graphene, it will be possible to actively control the high Q-factor EIT based on the toroidal dipole in all-dielectric metasurfaces. While the electromagnetic fields in all-dielectric metamaterials are generally confined within the resonators due to the Mie resonance properties of dielectric particles, the interaction between graphene and the electric field inside of the dielectric metamaterials is severely limited.

In this paper, we propose an all-dielectric metasurface composed of periodically arranged split Si nanocuboids that supports the high Q-factor EIT behavior in the optical telecommunication range. The high Q-factor EIT is excited due to the destructive interference between the toroidal dipole resonance and the magnetic dipole resonance. The large electric field enhancement is obtained in the split gap. With the integration of graphene, the excellent performance of an optical switch is realized in the EIT window by tuning the Fermi level of graphene, accompanied by the active control of the slow-light effect.

## 2. Nanostructure and Calculation Method

In brief, our proposed all-dielectric metasurface is composed of a periodically arranged split Si nanocuboid with a gap in the middle, placed over a silica substrate, as shown in Figure 1a. Figure 1b shows that the geometric parameters of the unit cell are *p* = 680 nm, *L* = 560 nm, *w* = 480 nm, *h* = 280 nm, and *g* = 90 nm. The optical constants of Si are taken from the measured data [32], and the refractive index of silica is supposed to be 1.45. The schematic diagram of the proposed hybrid graphene–dielectric metasurface is shown in Figure 1c, where the monolayer graphene is placed over the split nanocuboid. The plane wave is of normal incidence with an electric field along the *y* direction. The all-dielectric nanostructures can be easily fabricated by using the top-down method [14]. The graphene layer grown from the chemical vapor deposition can be transferred over the all-dielectric metasurfaces using standard transfer techniques [33]. The frequency-domain finite element method [34] is employed to investigate the optical properties of the metasurfaces. 

## 3. Properties of Graphene

The properties of graphene can be modulated in the infrared and optical frequency ranges by electrically tuning the Fermi energy. The optical conductivity of graphene can be well derived with the random-phase approximation theory in the local limit. It originates from the contributions of intraband and interband transitions as follows [35]:(1)$$\begin{array}{l} \sigma \left(\omega \right)=\frac{2e^{2}k_{B}T}{\pi \hbar^{2}}\frac{i}{\omega +i\tau^{-1}}\ln\left[2\cosh\left(\frac{E_{F}}{2k_{B}T}\right)\right]\\ \;\;\;+\frac{e^{2}}{4\hbar }\left[\frac{1}{2}+\frac{1}{\pi }\arctan\left(\frac{\hbar \omega -2E_{F}}{2k_{B}T}\right)-\frac{i}{2\pi }\ln\frac{{\left(\hbar \omega +2E_{F}\right)}^{2}}{{\left(\hbar \omega -2E_{F}\right)}^{2}+{\left(2k_{B}T\right)}^{2}}\right] \end{array}$$
where *e* is the electron charge, *k_B_* is the Boltzmann’s constant, *ћ* is the reduced Planck’s constant, *ω* is the angular frequency, *T* is the temperature and assumed to be 300 K, and *E_F_* is the Fermi energy (chemical potential). The relaxation time is represented by $\tau =\mu E_{F}/{\left(ev_{F}\right)}^{2}$, where $v_{F}\approx 10^{6}$ m/s is the Fermi velocity, and *μ* is the carrier mobility chosen as 10^4^ cm^2^/(V∙s). Figure 2a,b present the variation of the real part and imaginary part of the graphene conductivity with the incident light wavelength and Fermi level, respectively. The interband transitions in graphene dominate at near-infrared frequencies. When the Fermi level is less than the Dirac point by half of the photon energy (*E_F_* < *ћω*/2), the incident photons are absorbed by graphene due to the interband absorption, and the real part of the graphene conductivity has a large value. However, when *E_F_* > *ћω*/2, the interband transitions are blocked because of Pauli’s exclusion principle. Here, the real part of the conductivity quickly decreases to near zero when the Fermi level exceeds the critical value, as shown in Figure 2a. Hence, graphene becomes almost transparent for *E_F_* > *ћω*/2.

## 4. Results and Discussions

### 4.1. EIT in the All-Dielectric Metasurface

We firstly investigated the optical properties of the all-dielectric metasurface without the presence of graphene, as shown in Figure 1a. Figure 3 gives the transmission spectrum which presents a typical EIT resonance. To clarify its origin, we simulated magnetic field and electric field profiles of the EIT resonance in the *x*–*z* plane at *y* = 0 nm and the *x*–*y* mid-plane, as shown in Figure 4a,b, respectively. The circular magnetic moment was perpendicular to the nanocuboid surface, which generated a toroidal dipole moment in the plane of the nanocuboid. Hence, the peak of the transmission spectrum corresponded to the toroidal resonance. Figure 4c,d present the electric field profiles in the *y*–*z* plane at *x* = 200 nm at the right dip of the EIT peak and at the EIT peak, respectively, which indicate that the EIT effect results from the destructive interference between the toroidal dipole resonance and the magnetic dipole resonance. In addition, Figure 4b shows that the in-plane electric field is strongly confined in the split gap. Therefore, the all-dielectric metasurface can provide a platform for the enhanced interaction between light and the surrounding medium.

To gain further insight into the origin of this EIT peak, the scattering properties of a single split Si nanocuboid in the array are investigated by decomposing the electric field inside the nanoparticle into Cartesian multipole moments [36,37]. The induced displacement current density inside the nanoparticle can be expressed as [36]:(2)$$\vec{J}=-i\omega \left(\varepsilon -1\right)\vec{E}\left(\vec{r}\right)$$
where $\vec{E}\left(\vec{r}\right)$ is the electric field, *ε* is the permittivity of the nanocuboid. The amplitudes of the electric and magnetic multipole moments and the toroidal dipole moment are calculated as [36].

electric dipole moment:(3)$$\vec{p}=\frac{1}{i\omega }\int \vec{J}\left(\vec{r}\right)d^{3}r$$
magnetic dipole moment:(4)$$\vec{m}=\frac{1}{ic}\int \vec{r}\times \vec{J}\left(\vec{r}\right)d^{3}r$$
toroidal dipole moment:
(5)$$\vec{T}=\frac{1}{10c}\int \left\{\left[\vec{r}\cdot \vec{J}\left(\vec{r}\right)\right]\vec{r}-2\left(\vec{r}\cdot \vec{r}\right)\vec{J}\left(\vec{r}\right)\right\}d^{3}r$$
electric quadrupole moment:
(6)$$Q_{\alpha \beta }=\frac{1}{2i\omega }\int \left\{r_{\alpha }J_{\beta }\left(\vec{r}\right)+r_{\beta }J_{\alpha }\left(\vec{r}\right)-\frac{2}{3}\left[\vec{r}\cdot \vec{J}\left(\vec{r}\right)\right]\right\}d^{3}r$$
magnetic quadrupole moment:(7)$$M_{\alpha \beta }=\frac{1}{3c}\int \left\{{\left[\vec{r}\times \vec{J}\left(\vec{r}\right)\right]}_{\alpha }r_{\beta }+{\left[\vec{r}\times \vec{J}\left(\vec{r}\right)\right]}_{\beta }r_{\alpha }\right\}d^{3}r$$

The decomposed far-field scattered powers of these multipole moments can be calculated by using the following formulas: $I_{p}=\frac{2\omega^{4}}{3c^{3}}{\left|\vec{p}\right|}^{2}$, $I_{m}=\frac{2\omega^{4}}{3c^{3}}{\left|\vec{m}\right|}^{2}$, $I_{T}=\frac{2\omega^{6}}{3c^{5}}{\left|\vec{T}\right|}^{2}$, $I_{Q}^{e}=\frac{\omega^{6}}{5c^{5}}\sum {\left|Q_{\alpha \beta }\right|}^{2}$ and $I_{Q}^{m}=\frac{\omega^{6}}{40c^{5}}\sum {\left|M_{\alpha \beta }\right|}^{2}$. Their normalized scattered powers are presented in Figure 5a. It is found that the toroidal dipole is the strongest contributor, which is about 3.1 times stronger than the magnetic quadrupole and much stronger than other multipoles (P, M and Q_e_) around the resonant wavelength. This illustrates that the other multipoles are effectively suppressed, leading to the excitation of the toroidal dipole in our proposed metasurface. Moreover, the magnetic dipole shows larger values than the other multipoles in the full wavelength range, except near the resonance position. This further manifests that the EIT peak arises from the destructive interference between the toroidal dipole resonance and the magnetic dipole resonance. Figure 5a also presents the reconstructed transmission near the EIT peak from the normalized scattered powers [36], which agrees well with the calculated one, as shown in Figure 5b.

The Q-factor of the metasurface is significant to characterize the performance of devices in practical applications. This can be obtained by fitting it with an analytical interference model of the extinction spectra $E\left(\omega \right)={\left|e\left(\omega \right)\right|}^{2}$ [38]:(8)$$e\left(\omega \right)=a_{r}+\sum_{j}\frac{b_{j}\Gamma_{j}e^{i\varphi_{j}}}{\omega -\omega_{j}+i\Gamma_{j}}$$
where *a_r_* is the background amplitude, *b_j_*, *ω_j_*, Г*_j_* and *φ_j_* represent amplitude, resonant angular frequency, damping and phase of the *j* different oscillators that denote the interfering modes. The fitted spectrum is fairly consistent with the simulated spectrum, as shown in Figure 3. Here, the fitting resonant frequency and radiative damping of the toroidal mode are *ω*_2_ = 226.912 THz and Г_2_ = 0.052 THz, respectively. The Q-factor (*Q* = *ω*_2_/Г_2_) of the EIT is up to 4364, which is because the toroidal dipole effectively suppresses the radiative losses.

### 4.2. Transmission Switch

To investigate the performance of the optical switch based on the toroidal dipole resonance, we calculate the transmission spectra, as shown in Figure 6a–c, and the electric field profiles, as shown in Figure 6d–f, of the hybrid graphene–dielectric metasurface for different Fermi levels (*E_F_* = 0 eV, 0.5 eV and 0.6 eV). It is found that the transparency window of the EIT resonance undergoes a prominent transition with the Fermi level. The EIT position, i.e., the resonant wavelength of the toroidal dipole, is located at 1322 nm (*ћω*/2 = 0.47 eV). Compared with the all-dielectric metasurface without graphene, the transmission at the EIT position decays to about 0.085 for *E_F_* = 0 eV, and the electric field almost disappears. Under such a circumstance, the in-plane electric field of the toroidal resonance is strongly coupled to the interband transitions of the graphene, leading to the degeneration of the transmission and the electric field. As shown in Figure 6b,e, the transmission at the EIT position is changed to 0.42 and the electric field becomes strong when *E_F_* = 0.5 eV. When *E_F_* = 0.6 eV, the transmission at the EIT position becomes 0.932 and the electric field is even stronger, which is very close to that without the graphene overlayer, as presented in Figure 6c,f. Due to the prohibition of interband transitions, the interband absorption of graphene declines once *E_F_* > *ћω*/2. In Figure 2a, the real part of the graphene conductivity dramatically decreases with the increase in *E_F_*, which further proves the results. As *E_F_* further increases from 0.6 eV to 0.7 eV, the transmission at the EIT position only increases by 0.026, about 0.958. It is worth noting that the modulation of the transmission compared with the all-dielectric metasurface in our hybrid metasurface observes an opposite trend to [39,40]. This is because the change in the real part of the complex conductivity with the Fermi level in the telecommunication range is opposite to that in the THz band for a fixed frequency. The strong tunability of the transmission spectrum is realized in our proposed hybrid metasurface on account of the exceptional characteristic properties of graphene combined with the strong coupling to the enhanced electric field in the split gap.

Moreover, it is difficult for the ultrathin graphene to affect the resonant wavelength of the toroidal dipole, therefore, the induced transparency position is not sensitive to changes in the Fermi level. It is noteworthy that graphene cannot break the distribution patterns of the electric field, which gives rise to only the degeneration of the electric field amplitude and the resonance strength. Figure 7a gives the transmission at the EIT position as a function of the Fermi level of graphene. The transmission at the EIT position exhibits a crucial behavior when *E_F_* is around 0.47 eV (*ћω*/2). The transmission is very small when *E_F_* < 0.47 eV. It increases rapidly when *E_F_* goes through 0.47eV, and approaches saturation when *E_F_* = 0.6 eV. The light green area in Figure 7a denotes the transition region, that is, an on/off process that can be manipulated with the Fermi level of graphene. Therefore, an optical switch can be operated by electrically tuning the graphene-loaded all-dielectric metasurface.

In order to present the performance of the transmission switch more clearly, we calculate the modulation depth and the on/off ratio as a function of the wavelength of the incident plane wave in the system of the hybrid graphene–dielectric metasurface. The modulation depth is defined as the absolute value of the transmission difference $\Delta T=|T(E_{F}=0.7\mathrm{eV})-T(E_{F}=0\mathrm{eV})|\times 100\%$. As shown in Figure 7b, it is up to 88% at the EIT resonance, which is improved to an appreciable extent on the basis of those reported [21,22,23,39]; meanwhile the Δ*T* in the full wavelength range, except the resonance position, is approaching zero. In addition, the on/off ratio (the transmission ratio between *E_F_* = 0.7 eV and 0 eV) reaches up to 12.2 at the EIT resonance, as presented in the inset of Figure 7b. These results demonstrate that our proposed metasurface is a suitable candidate for an optical switch, an optical modulator, and an optical filter.

### 4.3. Modulation of Slow Light

The narrow transparency window of the EIT is always accompanied with strong dispersion behaviors, resulting in low group velocity of light propagation. The slow-light metamaterial can trap photons for a long time inside the structure, which is useful to enhance light–matter interactions, optical data processing, and optical switching [41]. The ability to actively control the slow light is investigated in our proposed metasurface. The group delay *τ_g_* and group index *n_g_* are introduced to describe the slow-light ability, which are expressed as [42]:(9)$$\tau_{g}=-\frac{d\phi }{d\omega }$$
(10)$$n_{g}=\frac{c}{v_{g}}=\frac{c}{h}\tau_{g}$$
where *c* is the speed of light in free space; *v_g_* is the group velocity; *h* is the thickness of the Si, and *ϕ* is the transmission phase. The group delay and index at the EIT resonance are calculated as a function of the Fermi level of graphene at the EIT resonance, as shown in Figure 8. The delay time is only about 1.0 ps when *E_F_* < 0.47 eV. As *E_F_* increases to 0.6 eV, the delay time is up to 3.1 ps. The delay time is comparable to the electron–phonon scattering time in graphene [43]. The *n_g_* is small, about 1120 when *E_F_* < 0.47 eV because the EIT effect is suppressed. When *E_F_* = 0.7 eV, the *n_g_* is up to 3390. Hence, the group velocity of light at the telecommunication wavelength can be tuned from *c*/1120 to *c*/3390 by modulating the Fermi level. This reveals that our proposed graphene-loaded all-dielectric metasurface provides an unprecedented opportunity for controllable slow-light devices.

## 5. Conclusions

In conclusion, the modulation of the EIT resonance is realized in our proposed graphene-loaded all-dielectric metasurface. The destructive interference between the toroidal dipole resonance and the magnetic dipole resonance results in the formation of the high Q-factor EIT in the all-dielectric metasurface. Meanwhile, the large electric field enhancement is obtained in the split gap. When graphene is covered on the all-dielectric metasurface, the transmission amplitude of the EIT resonance can be efficiently tuned by changing the Fermi level. The modulation depth and the on/off ratio are up to 88% and 12.2, respectively. Moreover, the group velocity can be tuned from *c*/1120 to *c*/3390. Our proposed hybrid metasurface exhibits a high modulation performance, opening up new opportunities for dynamic metadevices, such as optical filters, modulators, switches, and slow-light devices in the optical telecommunication range.

## Figures and Tables

**Figure 1 nanomaterials-10-01064-f001:**
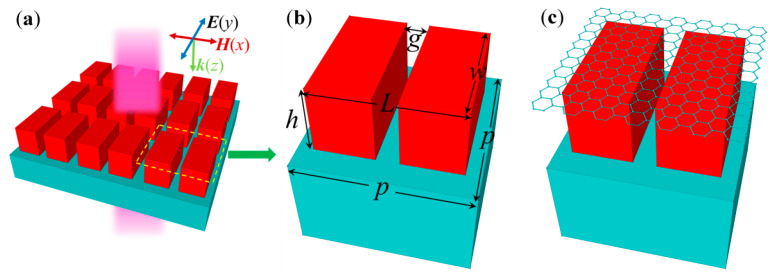
(**a**) Schematic of the all-dielectric metasurface. Diagrams of the unit cell for (**b**) the all-dielectric metasurface and (**c**) the graphene-loaded all-dielectric metasurface.

**Figure 2 nanomaterials-10-01064-f002:**
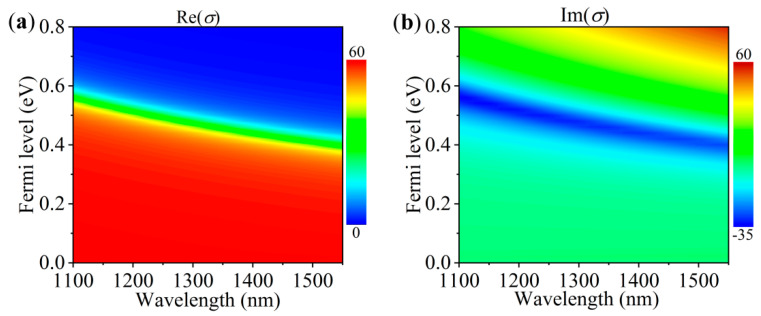
(**a**) Real and (**b**) imaginary part of the complex conductivity of graphene as a function of the incident light wavelength and the Fermi level.

**Figure 3 nanomaterials-10-01064-f003:**
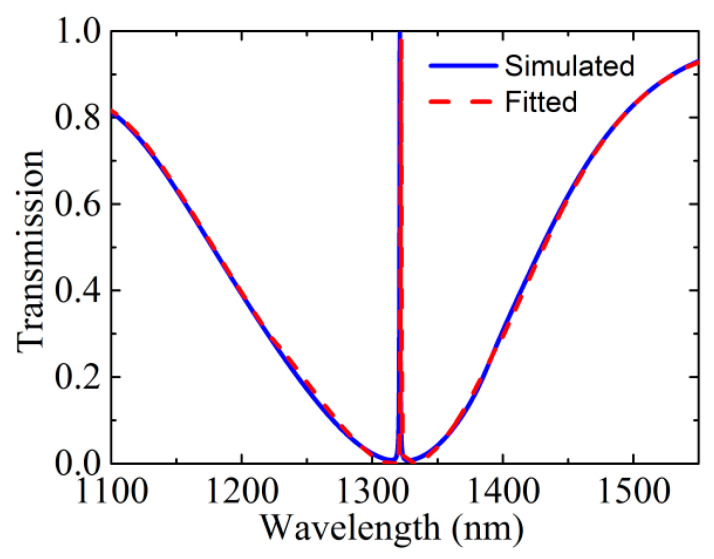
Transmission spectrum of the all-dielectric metasurface without graphene: the blue solid and the red dashed lines represent the simulated and fitted spectra, respectively.

**Figure 4 nanomaterials-10-01064-f004:**
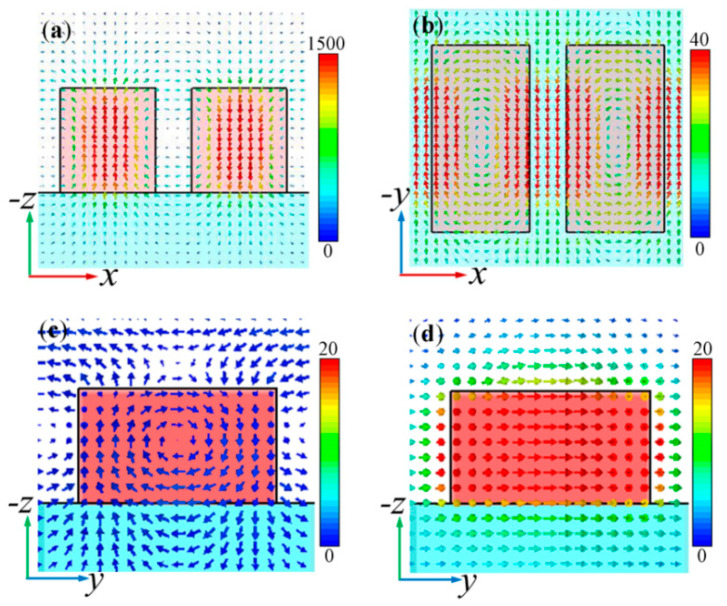
(**a**) Magnetic field profile in the *x*–*z* plane at *y* = 0 nm, and (**b**) electric field profile in the *x*–*y* mid-plane at the EIT peak. (**c**) and (**d**) correspond to electric field profiles in the *y*–*z* plane at *x* = 200 nm at the right dip of the EIT and at the EIT peak, respectively. The color scales represent the corresponding field enhancement.

**Figure 5 nanomaterials-10-01064-f005:**
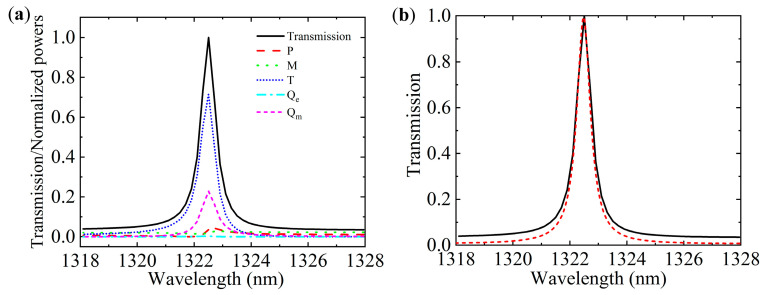
(**a**) Normalized scattered power of the five major multipoles and the reconstructed transmission of the all-dielectric metasurface. The five major multipoles are electric dipole (*P*), magnetic dipole (*M*), toroidal dipole (*T*), electric quadrupole (*Q*_e_), and magnetic quadrupole (*Q*_m_), respectively. (**b**) Calculated transmission (red dashed line) and reconstructed transmission (black solid line).

**Figure 6 nanomaterials-10-01064-f006:**
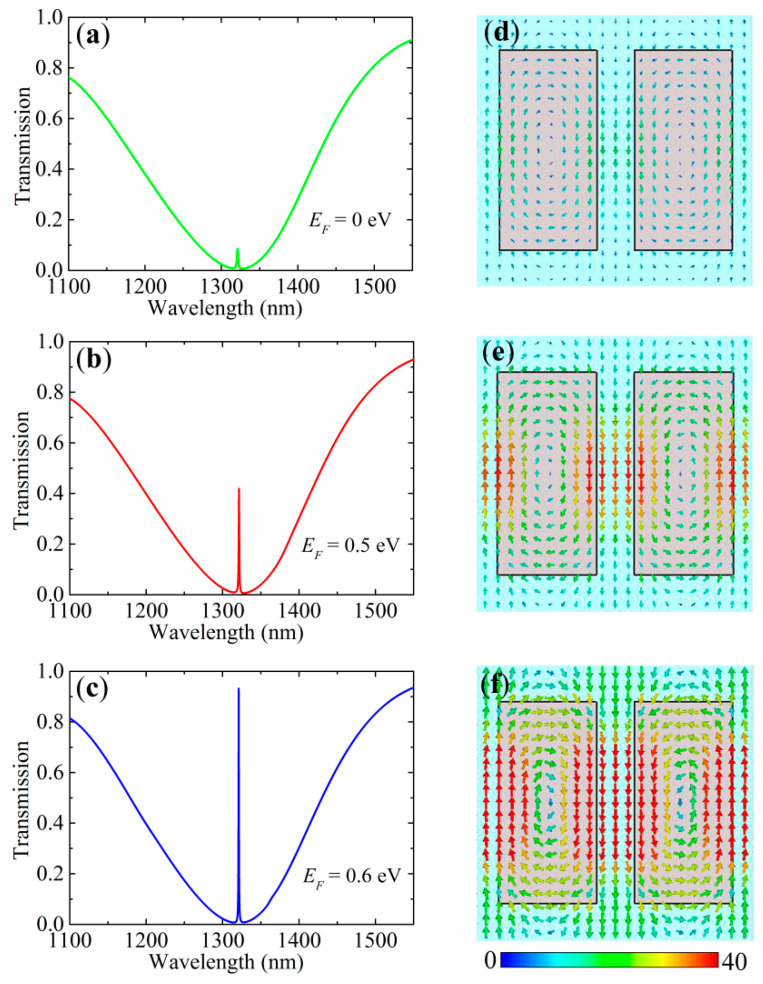
(**a**–**c**) Transmission spectra and (**d**–**f**) corresponding electric field distributions at the toroidal dipole resonance of the graphene-loaded all-dielectric metasurface for *E_F_* = 0 eV, 0.5 eV and 0.6 eV, respectively.

**Figure 7 nanomaterials-10-01064-f007:**
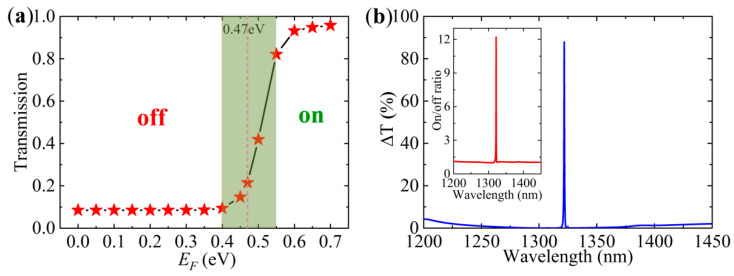
(**a**) The transmission at the EIT resonance as a function of the Fermi level. (**b**) Absolute value of the transmission difference as a function of the wavelength of the incident plane wave, and the inset is the on/off ratio.

**Figure 8 nanomaterials-10-01064-f008:**
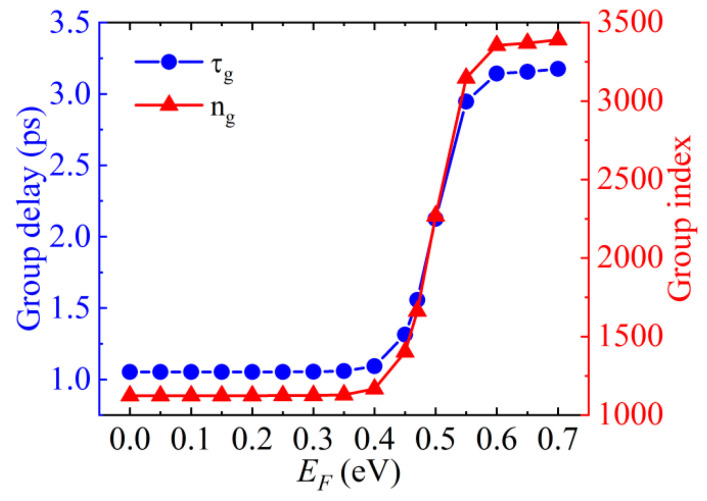
Group delay and index at the EIT resonance as a function of the Fermi level.

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
