# Peer review of "Switchable Electromagnetically Induced Transparency with Toroidal Mode in a Graphene-Loaded All-Dielectric Metasurface"

_nanomaterials, 2020, doi:10.3390/nano10061064_

Round 1

Reviewer 1 Report

In this manuscript, the authors report purely simulation results on
an all-electric metasurface. They mainly analyzed a narrow transmission
band in a telecom range. Toroidal moment was numerically evaluated,
following ref. 33.

A new twist is to include graphene as an electric controlling source.
On the transmission switch, it is unclear that how the complex conductivity
is related to the Fermi level E_F. This point should be explicitly described.
I think that the authors can write the equation and/or show diagram of
E_F vs \sigma. Also, they should address how to control E_F in a
practical configuration. Simple numerical results will not attract much
interest by readers.

To answer practical interest, it is preferred that they address
the switching speed (or the operation frequency).

Thus, this manuscript has a room to improve the significance;
it is not recommended to be published in the present form.

Reviewer 2 Report

The authors study theoretically the optical response of graphene-loaded dielectric metasurfaces. They claim that they observed EIT-like response and are able to tune it with the help of the graphene layer. Overall the study is correct and technically sound. But, there are some misconceptions and false statements. First, the authors mentioned that toroidal modes do no radiate - this is incorrect. They do radiate, but they can interfere with the electric dipole and result in anapole states, non-radiating configurations. Second, on Fig.4 they show the "scattering contribution" but they study periodic structures. Scattering contributions are only applicable to finite objects. Instead, they should calculate the contribution to the transmission, which is done in a different way. Third, in the same figure, they don't take into account the interference effect, which might diminish the strong toroidal dipole response. This should be corrected as well. It also implies that magnetic quadrupole will become dominant as well, so no pure toroidal dipole response. Thus, the title should be also adjusted.

Round 2

Reviewer 1 Report

The authors revised the manuscript, responding to the comments by the
reviewers. The purely numerical results are potentially applicable to
an optical modulation. But they do not at all show any possibility for
sensing performance of the metasurface; therefore, they should delete
the word "sensors" in the abstract. After this minor revision, this manuscript
could be publishable in Nanomaterials.

Reviewer 2 Report

The authors did not take into account all my comment. They insist on using single-cell scattering properties, which are not representative for the periodic structures and might be misleading. Thus, the proper contribution to the transmission/reflection should be demonstrated, not the scattering contributions. Second, it is irrelevant in the current study what is excited inside the particle, only their contributions to the transmission/reflection. And the authors should demonstrate that.

Round 3

Reviewer 2 Report

I would like to thank the authors to prove the detailed response. It now can be accepted for publication in the current form.
